# Apoptotic Effects of Anthocyanins from *Vitis coignetiae Pulliat* Are Enhanced by Augmented Enhancer of the Rudimentary Homolog (ERH) in Human Gastric Carcinoma MKN28 Cells

**DOI:** 10.3390/ijms22063030

**Published:** 2021-03-16

**Authors:** Cheol Park, Won Sup Lee, Se-Il Go, Sang-Ho Jeong, Jiyun Yoo, Hee-Jae Cha, Young-Joon Lee, Heui-Soo Kim, Sun-Hee Leem, Hye Jung Kim, Gon Sup Kim, Soon-Chan Hong, Yung Hyun Choi

**Affiliations:** 1Division of Basic Sciences, College of Liberal Studies, Dong-eui University, Busan 47340, Korea; parkch@deu.ac.kr; 2Departments of Internal Medicine, Institute of Health Sciences, Gyeongsang National University School of Medicine, Jinju 660-702, Korea; gose1@hanmail.net; 3Departments of Surgery, Institute of Health Sciences, Gyeongsang National University School of Medicine, Jinju 660-702, Korea; shjeong@gnu.ac.kr (S.-H.J.); orangejulia@naver.com (Y.-J.L.); hongsc@gnu.ac.kr (S.-C.H.); 4Department of Microbiology/Research Institute of Life Science, College of Natural Sciences, Jinju 660-701, Korea; yooj@gsnu.ac.kr; 5Department of Parasitology and Genetics, Kosin University College of Medicine, Busan 49267, Korea; hcha@kosin.ac.kr; 6Department of Biological Sciences, College of Natural Sciences, Pusan National University, Busan 46241, Korea; khs307@pusan.ac.kr; 7Departments of Biology and Biomedical Science, Dong-A University, Busan 49315, Korea; shleem@dau.ac.kr; 8Departments of Pharmacology, Institute of Health Sciences, Gyeongsang National University School of Medicine, Jinju 660-702, Korea; hyejungkim@gnu.ac.kr; 9School of Veterinary Medicine, Division of Applied Life Science (BK 21 Program), Gyeongsang National University, Jinju 660-701, Korea; gonskim@gnu.ac.kr; 10Department of Biochemistry, Dong-eui University College of Korean Medicine, Busan 47227, Korea

**Keywords:** anthocyanins, *Vitis coignetiae Pulliat*, apoptosis, MKN28 human gastric carcinoma cells, enhancer of the rudimentary homolog, anticancer effects

## Abstract

Evidence suggests that augmented expression of a certain gene can influence the efficacy of targeted and conventional chemotherapies. Here, we tested whether the high expression of enhancer of the rudimentary homolog (ERH), which serves as a prognostic factor in some cancers, can influence the efficacy of anthocyanins isolated from fruits of *Vitis coignetiae Pulliat*, Meoru in Korea (AIMs) on human gastric cancer cells. The anticancer efficacy of AIMs was augmented in ERH-transfected MKN28 cells (E-MKN28 cells). Molecularly, ERH augmented AIM-induced caspase-dependent apoptosis by activating caspase-3 and -9. The ERH-augmented apoptotic effect was related to mitochondrial depolarization and inhibition of antiapoptotic proteins, XIAP, and Bcl-2. In addition, reactive oxygen species (ROS) generation was augmented in AIMs-treated E-MKN28 cells compared to AIMs-treated naïve MKN28 cells. In conclusion, ERH augmented AIM-induced caspase-dependent mitochondrial-related apoptosis in MKN28 cells. A decrease in expression of Bcl-2 and subsequent excessive ROS generation would be the mechanism for ERH-augmented mitochondrial-related apoptosis in AIMs-treated MKN28 cells. A decrease in expression of XIAP would be another mechanism for ERH-augmented caspase-dependent apoptosis in AIMs-treated MKN28 cells.

## 1. Introduction

Gastric cancer (GC) is the third leading cause of cancer death worldwide [1] and the second cause of cancer death in Korea [2,3]. With medical advances, the survival outcome of operable GC has been improving [4,5]. However, for metastatic GC, a combination of traditional cytotoxic chemotherapies still showed limited results with a median survival of 9–11 months. Therefore, more intensive chemotherapy is needed, but it would increase toxicities which many patients cannot tolerate. In addition, with advances in medical science, the population of the elderly with cancer is increasing, and these patients cannot tolerate even conventional therapy with reduced doses [6,7]. For these reasons, targeted therapies have been adopted for the treatment of GC. Targeted therapies started with TOGA trial of trastuzumab (a monoclonal antibody against human epidermal growth factor receptor 2, HER2) in GC [8]. Ramucirumab targeting the extracellular domain of the vascular endothelial growth factor receptor (VEGFR) shows anticancer activities on advanced gastric or gastro-esophageal junction adenocarcinoma [9]. However, the anticancer effects of targeted therapy could be diminished or enhanced by activated signaling associated with cancer survival or cancer death [10]. In addition, the anticancer effect of a conventional chemotherapeutic agent, cisplatin, can be influenced by tumor suppressor p53 activation [11], and other chemotherapeutic agents can be, too [12]. Hence, customized therapies for cancer specific genotype get popular.

Recently, the population of the elderly with cancer has been increased due to medical advances, so the need for less toxic therapies has been increasing. The epidemiological study suggested that the high intake of colorful fruit and vegetables may reduce the incidence of cancer [13,14]. In addition, phytochemicals from natural herbs induce cancer cell death without showing any noticeable toxicities by safely modulating cancer cell biology [15]. Therefore, large attention has been paid to natural herbs or phytotherapy as an alternative therapy for cancer prevention or control.

*Vitis coignetiae* Pulliat (Meoru in Korea) traditionally has been used in Korean folk medicine for the treatment of inflammatory lesions or cancers. It contains an abundance of anthocyanins, the anticancer activities of which have been demonstrated in vitro and in vivo regarding apoptosis, cancer invasion, and angiogenesis [16,17,18,19,20]. In addition, the anthocyanins isolated from Meoru (AIMs) also exert various anticancer effects on cell death, proliferation, invasion, epithelial–mesenchymal transition (EMT), and metastasis [21,22,23,24,25]. The major mechanism of the anticancer effects of AIMs regarding cell death was apoptosis showing a distinctive phenotype: cytoplasmic blebbing, nuclear condensation and fragmentation, apoptotic bodies, and DNA fragmentation. However, little is known about whether the apoptotic effects of AIMs can be diminished when cancer cells are of aggressive features (highly proliferative and/or invasive features) by activating a certain gene.

Gastric cancer biology can be altered by its mutated or amplified signaling. Our colleagues found 430 proteins that are differentially expressed in gastric cancer tissues compared to normal tissues [26]. Among the 430 proteins, the enhancer of rudimentary homolog (ERH) gene is highly upregulated in gastric cancer tissue [26]. Although the role of ERH has not been clearly documented in cancers, Human Protein Atlas (https://www.proteinatlas.org/ENSG00000100632-ERH/pathology, accessed on 25 February 2021) suggests that ERH serves as a prognostic factor in liver, head and neck, breast, ovary, renal cell, gastric cancer, and lung cancer. However, the role is reversed depending on the cancer types; high expression of ERH serves as a good prognostic factor in breast, ovary and gastric cancer while it does as a poor prognostic factors in liver, head and neck, and prostate cancer. In addition, low ERH expression is associated with better survival in cancer patients whose cancers harbor KRAS mutations [27] while ERH was reportedly involved in cancer metastasis and EMT of bladder cancer [28]. Here, we investigated the effects of upregulated ERH gene on anticancer efficacies of AIMs in terms of apoptosis by transfecting ERH gene into MKN28 human gastric cancer cells.

## 2. Materials and Methods

### 2.1. Cells and Reagents

The MKN28 human gastric carcinoma cells obtained from the American type culture collection (Manassas, VA, USA) were cultured in RPMI medium (Invitrogen Corp, Carlsbad, CA, USA) supplemented with 10% (*v*/*v*) fetal bovine serum (FBS) (GIBCO BRL, Grand Island, NY, USA), 1 mM L-glutamine, 100 U/mL penicillin, and 100 μg/mL streptomycin at 37 °C in a humidified atmosphere of 95% air and 5% CO_2_. Antibodies against TRAIL, DR4, DR5, Fas, FasL, XIAP, a cellular inhibitor of apoptosis protein-1 (cIAP-1), cIAP-2, caspase-3, -8, -9, Bcl-2, Bax, Flip_L_, Flip_S_, PARP, β-catenin, and PLC**γ1** were purchased from Santa Cruz Biotechnology (Dallas, Texas, USA). An antibody against β-actin was from Sigma (Beverly, MA, USA). Caspase activity assay kits were obtained from R&D Systems (Minneapolis, MN, USA). An enhanced chemiluminescence (ECL) kit was purchased from Amersham (Arlington Heights, IL, USA). 5,5′,6,6′-tetrachloro-1,1′,3,3′-tetraethylbenzimidazolylcarbocyanine iodide (JC-1) were obtained from Calbiochem (San Diego, CA, USA). All other chemicals not specifically cited here were purchased from Sigma–Aldrich (St. Louis, MO, USA). All these solutions were stored at −20 °C. Stock solutions of 4,6-diamidino-2-phenylindole (DAPI, 100 μg/mL) and propidium iodide (PI, 1 mg/mL) were prepared in phosphate-buffered saline (PBS).

### 2.2. Anthocyanin Preparations

AIM was extracted from the fruits of Meoru. The well matured Meoru fruits were collected at Jiri Mountain, Republic of Korea. Purification and characterization of AIM (anthocyanins in Meoru) have been described previously [29]. AIMs consist of delphinidin-3,5-diglucoside (1): cyanidin-3,5-diglucoside (2): petunidin-3,5-diglucoside (3): delphinidin-3-glucoside (4): malvdin-3,5-diglucoside (5): peonidin-3,5-diglucoside (6): cyanidin-3-glucoside (7): petunidin-3-glucoside (8): peonidin-3-glucoside (9): malvidin-3- glucoside (10) = 3.5: 3.4: 7.1: 23.9: 8.0: 9.6: 9.1: 16.1: 5.7: 13.4.

### 2.3. Construction of the ERH Expression Plasmid and Transfection

Human ERH cDNA was purchased by OriGene (RC200367). Cells were transfected with ERH-expressing plasmids using Lipofectamine™ 3000 (Invitrogen; Thermo Fisher Scientific, Inc) as per the manufacturer’s instructions. After 48 h, the cells were treated with neomycin prior to selection. ERH expression in neomycin-resistant clones was examined by immunoblotting. We used two ERH-transfected stable MKN28 cell lines (E-MKN28 cells) in this study. For the two eoERH#5“ and eoERH#27“ stable cell lines, they were selected in the process of making a stable cell line, and the ones with the most expression level were selected. Mock-treated MKN28 cells (M-MKN 28 cells) refer to a cell that has been transfected with a control plasmid without the ERH gene. The pCMV6 empty vector was used to control cells.

### 2.4. Cell Viability Assay

The cell viability assay was measured by using a 3-(4, 5-dimethylthiazol-2-yl)-2, 5-diphenyltetrazolium bromide (MTT) assay and trypan blue exclusion methods. For the MTT assay, E-MKN28 cells and M-MKN28 cells were seeded at 10 × 10^4^ cells/mL in a 12-well plate. When a confluence of cells reached approximately 70%, the cells were treated with AIMs and incubated at indicated concentrations (up to 400 μg/mL) for 48 h, and then incubated in 0.5 mg/mL MTT solution for 3 h at 37 °C in the dark. The absorbance of each well was measured at 540 nm with an enzyme-linked immunosorbent assay (ELISA) reader (Sunnyvale, CA, USA). For the trypan blue exclusion methods, the supernatant was discarded, and the pellet was resuspended with culture medium. The cell suspension was mixed with trypan blue dye (Sigma-Aldrich Chemical Co., St. Louis, MO, USA), transferred to a hemocytometer, and live cells were counted under a phase-contrast microscope (Carl Zeiss, Oberkochen, Germany).

### 2.5. Flow Cytometry Analysis for Cell Cycle Analysis and Apoptosis

After treatment with AIMs at the concentrations of 200 μg/mL for 48 h, the MKN28 cells were collected, washed with cold PBS, and then centrifuged. The pellet was fixed in 75% (*v*/*v*) ethanol for 1 h at 4 °C. The cells were washed once with PBS and resuspended in cold PI solution (50 μg/mL) containing RNase A (0.1 mg/mL) in PBS (pH 7.4) for 30 min in the dark. For apoptosis analysis, the cells were collected, washed with ice cold PBS, and then centrifuged. Next, 5 μL of the annexin V conjugate was added to each 100 μL of cell suspension for 15 min, then 400 μL of annexin V-binding buffer was added, mixed gently, and the samples were kept on ice. Flow cytometry analyses were performed with Beckman Coulter cytomics FC 500 (San Jose, CA, USA).

### 2.6. Western Blot Analysis

After treatment with AIMs at the concentrations of 200 μg/mL for 48 h, MKN28 cells were harvested and lysed. Their proteins were quantified using the BioRad protein assay (Hercules, CA, USA). The proteins of the extracts were resolved by electrophoresis, electrotransferred to a polyvinylidene difluoride membrane from Millipore (Bedford, MA, USA), and then the membrane was incubated with the primary antibodies followed by a conjugated secondary antibody to peroxidase. Blots were developed under enhanced chemiluminescence (ECL) detection system (Amersham).

### 2.7. In Vitro Caspases Activity Assay

Caspases activity was measured in MKN28 cells after treated with AIMs at the concentrations of 200 μg/mL for 48 h using by colorimetric assay kits, which utilized the following synthetic tetra-peptides, labeled with p-nitroaniline (pNA): Asp-Glu-Val-Asp (DEAD) for caspase-3, Ile-Glu-Thr-Asp (IETD) for caspase-8, and Leu-Glu-His-Asp (LEHD) for caspase-9. The cells were lysed in the supplied lysis buffer. The supernatants were collected and incubated with the supplied reaction buffer containing dithiothreitol and substrates at 37 °C. The caspase activities were determined by absorbance at 405 nm using the microplate reader.

### 2.8. Measurement of Mitochondrial Membrane Potential (MMP, Δψm) and ROS Generation

MKN28 cells were treated with AIMs at the concentrations of 200 μg/mL for 48 h and MMP (*ΔΨm*) in living cells was measured by flow cytometry with the lipophilic cationic probe JC-1, a ratiometric, dual-emission fluorescent dye. There are two excitation wavelengths, 527 nm (green) for the monomer form and 590 nm (red) for the J-aggregate form. The cells were harvested and resuspended in 500 μL of PBS, incubated with 10 μM JC-1 for 20 min at 37 °C. The quantitation of green fluorescent signals reflects the amounts of damaged mitochondria. For ROS measurement, the cells were incubated with 10 μM 2,7 -dichlorofluorescein diacetate (DCF-DA) at 37 °C for 30 min. The cells were then washed with ice-cold PBS and harvested. Fluorescence was determined by a FACS flow cytometer.

### 2.9. Statistical Analysis

Each experiment was performed in triplicate. The results were expressed as means ± SD. Significant differences were determined using the one-way analysis of variance (ANOVA) with post-test Neuman–Keuls for the cases at least three treatment groups and Student’s *t*-test for two-group comparison. Statistical significance was defined as *p* < 0.05.

## 3. Results

### 3.1. ERH Augmented The Antiproliferative Effect of Aims on MKN28 Cells

To investigate the effects of an augmented expression of ERH on anticancer activity of AIMs, we treated ERH-transfected MKN28 cells (E-MKN28 cells) and mock-treated MKN28 cells (M-MKN28 cells) with AIMs at indicated concentrations (up to 400 μg/mL) for 48 h and assessed the effects of ERH on AIM-treated cells with trypan blue exclusion test and MTT assay. Trypan blue exclusion test and MTT assay both revealed that the growth of E-MKN28 cells was more inhibited by AIMs than that of M-MKN28 cells (Figure 1A,B). The efficacy of ERH gene on AIM-induced antiproliferative activity was greatest at the concentration of 200 μg/mL of AIMs. These results clearly indicated that ERH augmented the antiproliferative effect of AIMs on MKN28 cells.

### 3.2. ERH Augmented AIM-Induced Apoptosis in MKN28 Cells

From the above experiments, we considered 200 μg/mL of AIMs as an optimal concentration for the experiment. Next, we performed PI/annexin V double staining analysis, DAPI staining, and DNA fragmentation test to determine whether the antiproliferative effect of ERH on AIM-treated cells was attributed to apoptosis. PI/annexin V double staining analysis indicated that ERH augmented the late phase apoptosis or necrosis effect of AIMs on MKN28 cells (Figure 2A). DAPI staining also indicated that ERH increased the apoptotic effects of AIMs on MKN28 cells (Figure 2B). DNA fragmentation test showed that a typical step ladder pattern of nuclear fragmentation, an indicator of apoptosis was observed in AIM-treated E-MKN28 cells (Figure 2C). These results indicated that ERH augmented the apoptotic effect of AIMs on MKN28 cells.

### 3.3. ERH Augmented AIM-Induced Caspase-Dependent Apoptosis Possibly Through the Intrinsic Apoptotic Pathway in MKN28 Cells

To determine whether the ERH-augmented apoptotic effect was caspase-dependent or not, we next performed Western blot analysis for caspases and caspase activity assay. Western blot analysis revealed that ERH augmented the activation of caspase-3 and -9 induced by AIMs, and a decrease in PARP, β-catenin, and PLCγ1 (nuclear proteins that can be cleaved by caspase 3) in AIMs-treated cells (Figure 3A). Caspase activity assay also showed that ERH augmented the activation of caspase-3 and -9 induced by AIMs (Figure 3B). As the clear cleavage of PARP was not observed, we confirmed by caspase-3 inhibitor assay that ERH-augmented apoptotic effect was caspase-dependent. As shown in Figure 4 (in MTT assay (Figure 4A), DNA fragmentation test (Figure 4B), DAPI staining (Figure 4C), and PI/annexin V double staining analysis (Figure 4D), Z-DEVD-FMK (a caspase-3 inhibitor) inhibited the ERH-augmented apoptotic effect of AIMs on MKN28 cells. These results strongly suggested that ERH augment the caspase-dependent apoptotic effect of AIMs on MKN28 cells probably through the intrinsic apoptotic pathway.

### 3.4. The ERH-Augmented Apoptotic Effect on AIM-Treated Cells Was Related to Mitochondrial Depolarization and Inhibition of Antiapoptotic Proteins, XIAP and Bcl-2

In the intrinsic apoptotic pathway, mitochondrial depolarization plays a central role [30]. Here, we measured the changes in MMP (*ΔΨ_m_*). As shown in Figure 5A, ERH augmented AIM-induced MMP (*ΔΨ_m_*) loss while ERH itself did not. Next, we conducted Western blot analysis for the death receptors, and IAP and Bcl-2 family members. The Western blot analysis revealed that ERH decreased the expression of antiapoptotic protein XIAP and Bcl-2 in both AIM-untreated and AIM-treated cells (Figure 5C,D). The results indicated that ERH-augmented apoptotic effect on AIM-treated cells was related to mitochondrial depolarization and inhibition of antiapoptotic proteins, XIAP, and Bcl-2.

### 3.5. ERH Augmented ROS Generation Triggered by AIMs

We next determined whether or not intracellular ROS generation contributed mitochondrial depolarization in apoptosis because ROS generation is a frequent mechanism for intrinsic apoptosis [30,31]. We measured ROS with DCF staining for 6 h. As shown in Figure 6, ERH itself induced ROS generation and augmented AIM-induced ROS generation after 30 min and the efficacy was decreased as time passed by. Then, with NAC treatment, we confirmed that ROS generation is involved in the ERH-augmented apoptosis in MTT assay (Figure 7A), DNA fragmentation test (Figure 7B), DAPI staining (Figure 7C), and PI/annexin V double staining analysis (Figure 7D). These results indicated that ROS generation was involved in the ERH-augmented apoptotic effect on AIM treated cells.

## 4. Discussion

The present study was designed to determine whether highly expressed ERH could influence the anticancer effects of AIMs on MKN28 human gastric cancer cells and to further explore the molecular mechanisms. This study demonstrated that increased expression of ERH augmented the anticancer effects of AIMs on MKN28 cells by intensifying the caspase-dependent intrinsic apoptosis through ROS generation and a decrease in expression of Bcl-2 and XIAP. All these three components can definitely influence on AIMs-induced caspase-dependent apoptosis, but each component plays a different role in ERH-augmented apoptosis in AIMs-treated MKN28 cells. Let us start discussing ROS first. Cancer cells can survive a high basal ROS level due to increased antioxidant production capacity [32]. Therefore, increased basal level of ROS that was induced by ERH may not serve as triggering signal for the apoptosis because E-MKN28 cells did not show any more cell death than M-MKN28 cells did (Figure 1). However, the high basal ROS level could influence the biological behavior of E-MKN28 cells, because ROS can serve as signaling molecules. As previously reported, high ROS can facilitate cancer metastasis [33,34,35]. However, our team found that E-MKN28 cells are less migratory and invasive than M-MKN28 cells, and inhibition of ERH of MKN28 cells by si-RNA led to an increase in migration and invasion. In addition, tissue microarray with 327 gastric tissue samples revealed that high expression of ERH in gastric cancer tissue serves as good prognostic factors (data not shown). ROS generation by ERH transfection did not serve as a stimulator of invasion or migration. Recent literature suggested that ROS serves as a dual effect depending on ROS generation amounts and antioxidant capacity. Actually, under certain conditions, high ROS increase proapoptotic molecules, or ROS levels above the toxic threshold can induce vulnerability to cell death, apoptosis, and senescence [36].

A decreased expression of Bcl-2 should also play a critical role in ERH-augmented apoptotic effects because Bcl-2 serves as a control switch for mitochondrial-related apoptosis [37]. As supporting evidence, the fate of cancer cells is greatly influenced by the expression of Bcl-2; many cancer cells harboring overexpression of Bcl-2 exhibited resistance to conventional chemotherapies that trigger apoptosis [38,39]. In addition, many natural polyphenols induced cancer cell death by downregulating Bcl-2 expression [40]. Therefore, the decreased expression of Bcl-2 in E-MKN28 cells contributes to making MKN28 cells more susceptible to mitochondrial membrane permeabilization (MOMP) that is crucial for mitochondrial-related apoptosis. Initially AIMs induces MOMP, and then large amounts of superoxide (a type of ROS in the mitochondria) from mitochondria should be released, which should lead to an imbalance of reduction-oxidation (Redox) system, and the excess ROS triggers more damage on mitochondria as well as other vital molecules [41]. Therefore, a decrease in Bcl-2 expression in E-MKN28 cells made E-MKN28 cells highly susceptible to MOMP and augmented mitochondrial-related apoptosis by ROS release from mitochondria. 

Lastly, XIAP is considered as the most potent caspase-binding protein because it inhibits caspases at both the initiation phase (caspase 9) and the final execution phase (caspase 3) of caspase-dependent apoptosis [42]. The decreased XIAP expression in E-MKN28 cells should make the cells highly susceptible to caspase-dependent apoptosis. As supporting evidence, the overexpression of XIAP in cancer is involved in resistance to conventional chemotherapies and targeted therapies [43,44,45]. Therefore, XIAP also played an important role in augmenting AIM-induced caspase-dependent apoptosis in E-MKN28 cells.

The weakness of this study is as follows: first, the role of ERH is not fully documented in GC. We tested whether ERH can enhance invasion and it inhibited cancer cell migration and invasion [46]. In addition, immunohistochemical staining with tissue microarray harboring 319 GC tissue samples showed lower cancer recurrence rates and longer survival times than patients tumors showing low expression of ERH [46]. This finding suggests that ERH may serve as a good prognostic factor. However, this result is opposite to previous reports in other cancers such as bladder, liver, and prostate [28] (https://www.proteinatlas.org/ENSG00000100632-ERH/pathology accessed on 1 February 2021). ERH may play a cancer-specific role among cancers. Regarding cancer invasion, and metastasis in GC, the role of ERH needs elucidating. 

Second, we did not investigate how ERH increased basal level of ROS and decreased Bcl-2 and XIAP expression. This question needs further in-depth research regarding the causal relationship and the detailed mechanisms for the reduction of Bcl-2 and XIAP. Even though in this study we did not investigate the mechanisms, we clearly demonstrated that ERH augmented caspase-dependent mitochondrial-related apoptosis of AIMs on MKN28 cells by excessive intracellular ROS production that was involved in a decrease in expression of Bcl-2 and subsequent excessive ROS release from mitochondria.

Third, we did not investigate whether the increased anticancer effect of AIM with the increase of ERH was related to the expression of multiple cell cycle and DNA damage response (DDR) genes that is deeply involved with ERH. As it was reported, ERH played a critical role in the mRNA splicing, the expression of the mitotic motor protein CENP-E, and consequently, chromosome congression [27]. To show the strong causal relationship between the overexpression of ERH and the increased anticancer activity of AIMs should be present. However, the role of ERH is not fully elucidated in cancer, and AIM did not affect the cell cycle and DNA damage. To perform this investigation, we should prepare a lot of things, so we will consider it in the next study. In the next study, we will consider testing the effects of AIMs on cell cycle, nuclear morphology, chromosome congression, and the expression of multiple cell cycle and DNA damage response (DDR) genes to evaluate whether AIMs could modulate cell viability by affecting ERH functions [27].

Fourth, cancer cells that harbor KRAS mutations are particularly sensitive to suppression of ERH. The ERH expression is inversely correlated with survival of cancer patients whose cancers harbor KRAS mutations [27], while it is not associated with survival in those patients whose cancers are wild-type KRAS or harbor other oncogenic mutations such as EGFR or PI3K mutations. It would be interesting to test the influence of ERH on the anticancer effects of AIMs. We would consider it in the next study, too.

The merit of this study is that we confirmed each finding with several other methods. For example, to confirm caspase-dependent apoptosis, we performed Western blot analysis, caspase activity assay, and caspase inhibitor assay.

In conclusion, these results indicated that ERH itself induced ROS generation and decreased the expression of Bcl-2 and XIAP and that ERH augmented caspase-dependent mitochondrial-related apoptosis of AIMs on MKN28 cells by excessive intracellular ROS production. Among the three components that were induced by ERH, a decrease in expression of Bcl-2 and subsequent excessive ROS generation triggered by AIM treatment would be the main mechanism for ERH-augmented mitochondrial-related apoptosis in AIM-treated MKN28 cells. A decrease in expression of XIAP would be a supportive mechanism for ERH-augmented caspase-dependent apoptosis in AIM-treated MKN28 cells. This study provides evidence that gene expression should influence phytotherapy for cancer treatment.

## Figures and Tables

**Figure 1 ijms-22-03030-f001:**
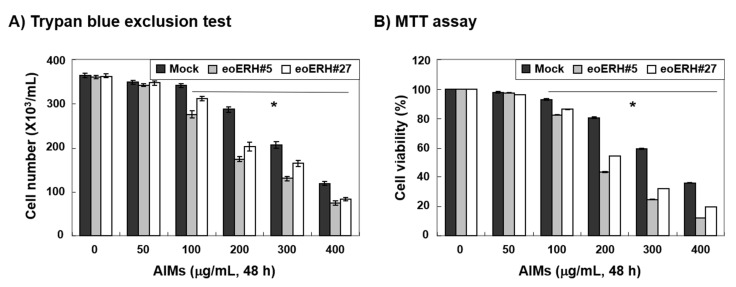
Enhancer of rudimentary homolog (ERH)-augmented antiproliferative effect on AIM-treated MKN28 cells. The cells were seeded at the density of 5 × 10^4^ cells per mL. The two kinds of ERH-transfected MKN28 cells (E-MKN28 cells: eoERH#5 and eoERH#27) and mock-treated MKN28 cells (M-MKN28 cells) were treated with AIMs at indicated concentrations (up to 400 μg/mL). The efficacy of ERH gene on AIM-induced antiproliferative activity was greatest at the concentration of 200 μg/mL of AIMs. The inhibition of cell proliferation was measured by (**A**) trypan blue exclusion test and (**B**) MTT assay. The data are shown as means ± SD of three independent experiments. ‘*’ represents significance (* *p* < 0.05 between AIM-treated E-MKN28 cells and AIM-treated M-MKN28 cells).

**Figure 2 ijms-22-03030-f002:**
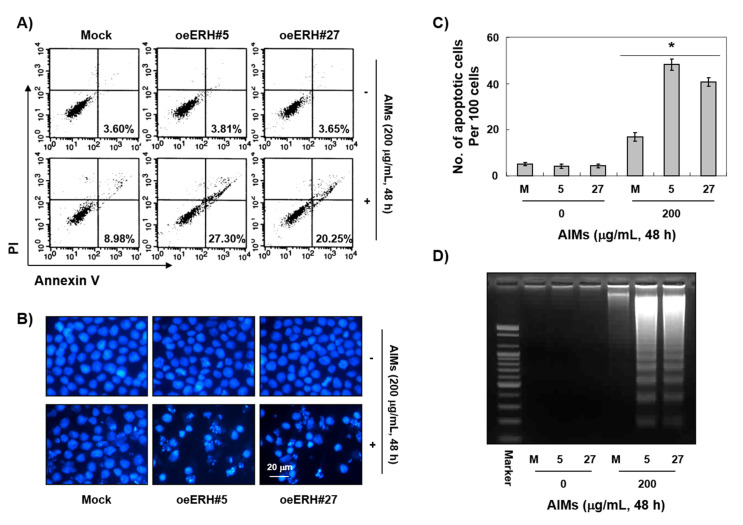
ERH-augmented apoptotic effects on AIM-treated MKN28 cells. The cells were seeded at the density of 5 × 10^4^ cells per mL. The two kinds of E-MKN28 cells (eoERH#5 and eoERH#27) and M-MKN28 cells were treated with AIMs at the concentrations of 200 μg/mL for 48 h. ERH increased the apoptotic effects of AIM on MKN28 cells. (**A**) Annexin V/PI flow cytometry assay, (**B**) nuclear staining with DAPI solution (Magnification, ×400), and (**C**) quantitative results for the number of apoptotic cells per 100 cells in total. * *p* < 0.05 vs. control group. (**D**) DNA fragmentation test. A ladder pattern of DNA fragmentation indicates internucleosomal cleavage associated with apoptosis. The data are representative of three independent experiments. M, M-MKN28 cells; 5, eoERH#5 E-MKN28 cells; 27, eoERH#27 E-MKN28 cells.

**Figure 3 ijms-22-03030-f003:**
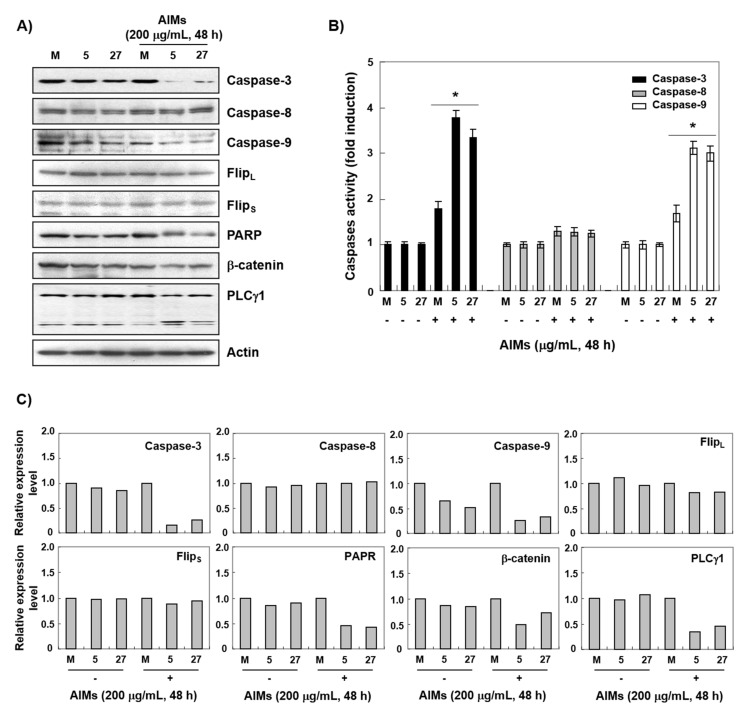
ERH-augmented activation of caspase-3 and -9 in AIM-treated MKN28 cells. The cells were seeded at the density of 5 × 10^4^ cells per mL. The two kinds of E-MKN28 cells (eoERH#5 and eoERH#27) and M-MKN28 cells were treated with AIMs at the concentrations of 200 μg/mL for 48 h. (**A**) Western blot analysis for procaspase-3, 8, 9, PARP, β-catenin, and PLCγ1 antibodies, (**B**) caspase activity assay, and (**C**) densitometry analysis of the data in Western blot analysis by ImageJ software. The values were normalized against β-actin. ERH augmented activation of caspase-3 and -9, in AIMs-treated cells. The data shown as figures are representative of three independent experiments. The data shown in the bar graph are of three independent experiments. ‘*’ represents significance (**p* < 0.05 between AIMs-treated E-MKN28 cells and AIMs-treated M-MKN28 cells). M, M-MKN28 cells; 5, eoERH#5 E-MKN28 cells; 27, eoERH#27 E-MKN28 cells.

**Figure 4 ijms-22-03030-f004:**
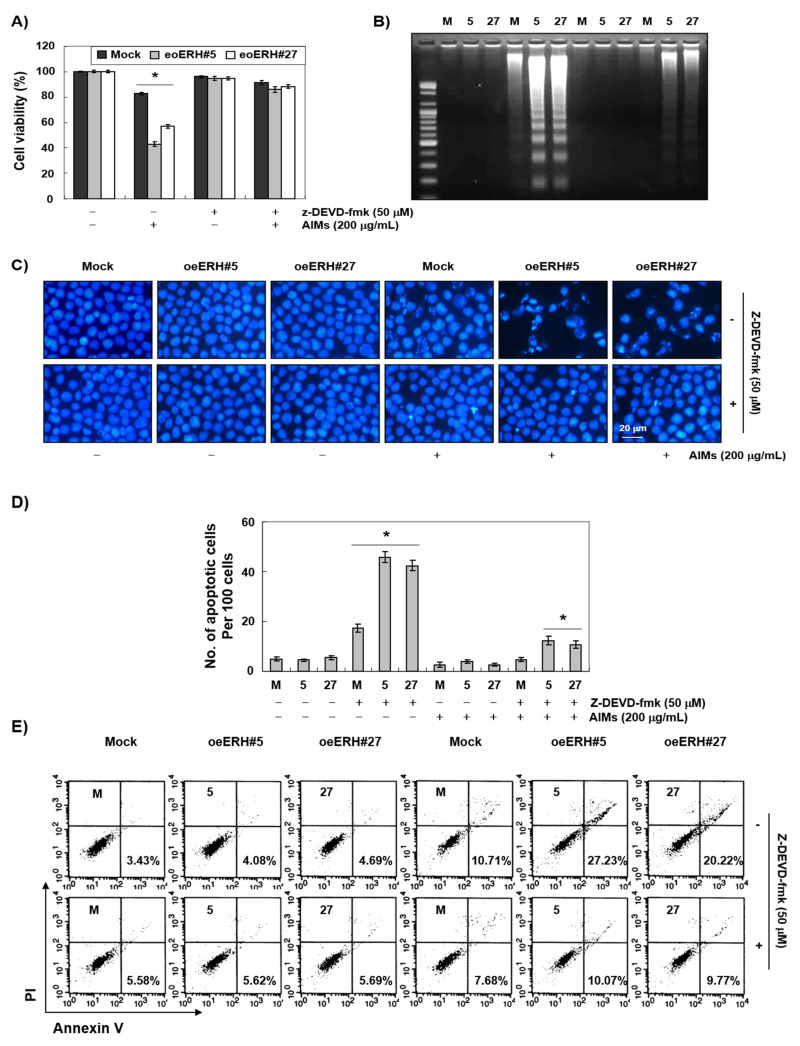
ERH-augmented caspase-dependent apoptosis in AIM-treated MKN28 cells. The cells were seeded at the density of 5 × 10^4^ cells per mL. The two kinds of E-MKN28 cells (eoERH#5 and eoERH#27) and M-MKN28 cells were treated with AIMs at the concentrations of 200 μg/mL for 48 h. Caspase 3 inhibitor (z-DEVD-fmk) inhibited the ERH-augmented effects on AIM-induced caspase-dependent apoptosis in (**A**) MTT assay, (**B**) DNA fragmentation test, (**C**) nuclear staining with DAPI solution (Magnification, ×400), (**D**) quantitative results for the number of apoptotic cells per 100 cells in total, and (**E**) Annexin V/PI flow cytometry assay. The data shown in the bar graph are of three independent experiments. ‘*’ represents significance (**p* < 0.05 between groups treated with AIMs alone and treated with AIMs combined with z-DEVD-fmk). The data shown as figures are representative of three independent experiments. M, M-MKN28 cells; 5, eoERH#5 E-MKN28 cells; 27, eoERH#27 E-MKN28 cells.

**Figure 5 ijms-22-03030-f005:**
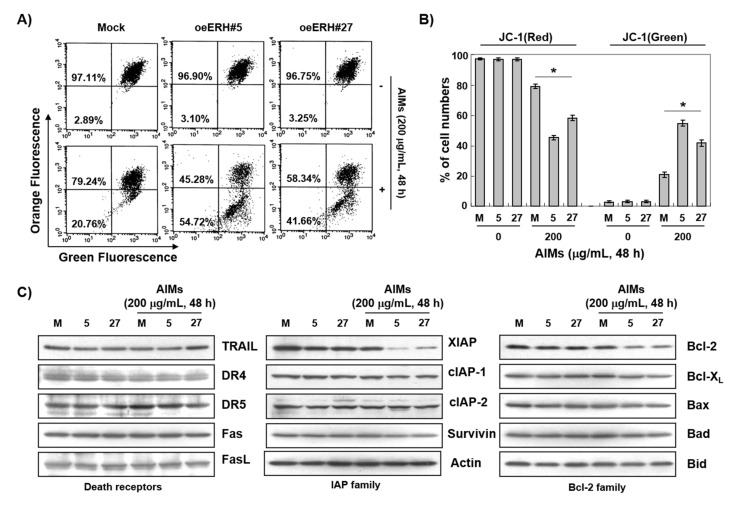
The mechanisms for ERH-augmented apoptotic effect on AIM-treated cells: mitochondrial depolarization and inhibition of Bcl-2 and XIAP expression. The cells were seeded at the density of 5 × 10^4^ cells per mL. The two kinds of E-MKN28 cells (eoERH#5 and eoERH#27) and M-MKN28 cells were treated with AIMs at the concentrations of 200 μg/mL for 48 h. (**A**) AIMs induced more loss of MMP (*ΔΨ_m_*) (mitochondrial depolarization) in E-MKN28 cells than M-MKN28 cells. (**B**) The data are shown as means ± SD of three independent experiments. ‘*’ represents significance (* *p* < 0.05 between AIM-treated E-MKN28 cells and AIM-treated M-MKN28 cells). (**C**) Western blot analysis, which revealed that ERH decreased the expression of antiapoptotic protein XIAP and Bcl-2 in both AIM-untreated and AIM-treated cells. The Western blot analysis data are representative of three independent experiments. (**D**) Densitometry analysis of the data in Western blot analysis by ImageJ software. The values were normalized against β-actin. (* *p* < 0.05 between AIM-treated E-MKN28 cells and AIM-treated M-MKN28 cells). M, M-MKN28 cells; 5, eoERH#5 E-MKN28 cells; 27, eoERH#27 E-MKN28 cells.

**Figure 6 ijms-22-03030-f006:**
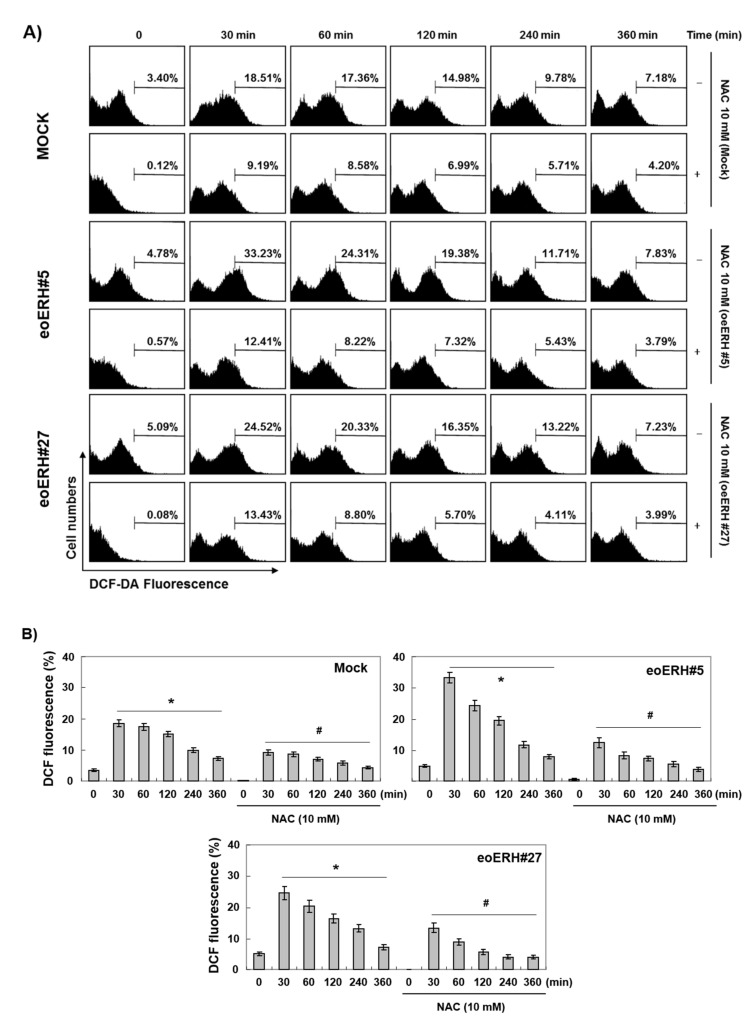
ERH-augmented ROS generation triggered by AIMs. The cells were seeded at the density of 5 × 10^4^ cells per mL. The two kinds of E-MKN28 cells (eoERH#5 and eoERH#27) and M-MKN28 cells were treated with AIMs at the concentration of 200 μg/mL for 48 h. ROS generation triggered by AIMs is the highest at 30 min after AIMs treatment. At this time, ROS generation is much higher in E-MKN28 cells than M-MKN28 cells. (**A**) The data shown in the figure are representative of three independent experiments. (**B**) Summarized flow cytometry data in bar graphs. (*, # *p* < 0.05 between before and after AIM treatment). M, M-MKN28 cells; 5, eoERH#5 E-MKN28 cells; 27, eoERH#27 E-MKN28 cells.

**Figure 7 ijms-22-03030-f007:**
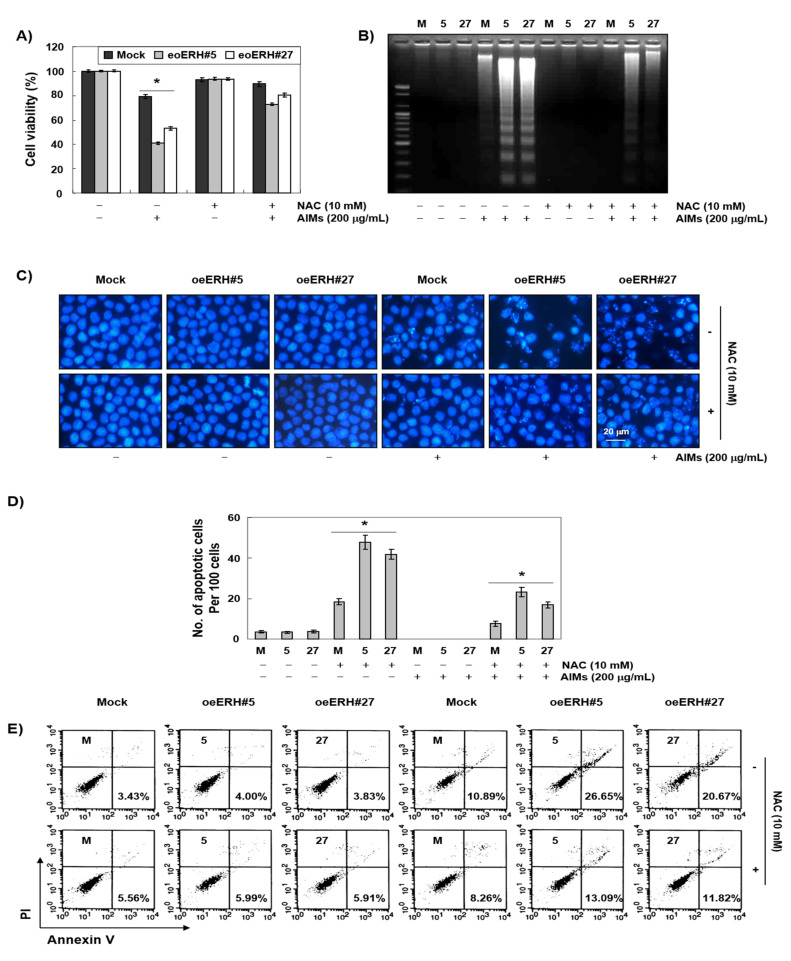
Another mechanism for ERH-augmented apoptotic effect on AIM-treated cells: excessive ROS generation triggered by AIMs. The cells were seeded at the density of 5 × 10^4^ cells per mL. The two kinds of E-MKN28 cells (eoERH#5 and eoERH#27) and M-MKN28 cells were treated with AIMs at the concentrations of 200 μg/mL for 48 h. With NAC treatment, the ERH-augmented apoptosis was diminished in (**A**) MTT assay, (**B**) DNA fragmentation test, (**C**) nuclear staining with DAPI solution (magnification, ×400), (**D**) quantitative results for the number of apoptotic cells per 100 cells in total, and (**E**) Annexin V/PI flow cytometry assay. The data are shown as means ± SD of three independent experiments. ‘*’ represents significance (* *p* < 0.05 between AIM-treated E-MKN28 cells and AIM-treated M-MKN28 cells). The data shown in figure are representative of three independent experiments. M, M-MKN28 cells; 5, eoERH#5 E-MKN28 cells; 27, eoERH#27 E-MKN28 cells.

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
