# Peer review of "Apoptotic Effects of Anthocyanins from Vitis coignetiae Pulliat Are Enhanced by Augmented Enhancer of the Rudimentary Homolog (ERH) in Human Gastric Carcinoma MKN28 Cells"

_ijms, 2021, doi:10.3390/ijms22063030_

Round 1
Reviewer 1 Report
Park et al. in his recent manuscript describe and influence of anthocyanins extracted from Vitis coignetiae Pulliat in gastric cancer cells MKN28. The manuscript presents some potentially interesting data. However, at least major revision (if not additional experiments) is required before acceptance in IJMS.
- Based on Human Protein Atlas (https://www.proteinatlas.org/ENSG00000100632-ERH/pathology) ERH has low cancer specificity. Why authors picked that particular gene/protein? The Ref. 26 in the manuscript refers to the publication that does not even mention ERH, and studies in Ref. 27 were done on bladder cancer cells. Was there any paritcular reason to pick MKN28 cell lines?
- Moreover, both introduction and materials and methods do not provide sufficient information about AIM's. Authors have to provide more detailed description of their composition (HPLC analysis?)
- In M&M authors suddenly refers to E-MKN28 cells and M-MKN28 cells without mentioning how to interpret these differences.
- Based on the description provided by the authors the Annexin V-based experiment was performed improperly. Annexin V conjugate should be added to the cells suspended in binding buffer, not in PBS (the provided description suggests that).
- Fig. 3 A and C - the axis description should be changed from caspase to pro-caspase. The present description is inaccurate and misleading.
6. How many biologically independent repeats was performed for each experiment? Whiskers of plots present SD? Some data presents almost ideal repeatability (WB results almost no discrepancies between samples?)
minor issues
line 74 - In vitro -> in vitro
line 220 - AS -> As
line 229, 241, 265, 289, 299 x -> ×
line 275 weird symbol in 'treated' word
Author Response
Park et al. in his recent manuscript describe and influence of anthocyanins extracted from Vitis coignetiae Pulliat in gastric cancer cells MKN28. The manuscript presents some potentially interesting data. However, at least major revision (if not additional experiments) is required before acceptance in IJMS.
Based on Human Protein Atlas (https://www.proteinatlas.org/ENSG00000100632-ERH/pathology) ERH has low cancer specificity. Why authors picked that particular gene/protein?
Thank you for your comments and a good question. We previously identify 430 candidate target proteins from 152 human Gastric cancer specimens that is differentially expressed in gastric tumor tissues compared to those in healthy tissues. Of the 430 proteins, enhancer of rudimentary homolog (ERH) was identified [1] [2]
In addition, ERH expression is also elevated in breast cancer (Zafrakas et al., 2008). ERH serves as a prognostic factor in liver, head and neck cancer, ovary and renal cell cancer.
With review of papers, we found that ERH was related to cancer metastasis and EMT of bladder cancer [3]. Therefore, we thought that investigation of ERH would be worthwhile.
The Ref. 26 in the manuscript refers to the publication that does not even mention ERH, and studies in Ref. 27 were done on bladder cancer cells. Was there any paritcular reason to pick MKN28 cell lines?
Thank you for your comments and a good question. Our team members previously identify 430 candidate target proteins from 152 human Gastric cancer specimens with publishing the paper [4], but the corresponding author did not want to release all the target proteins. I feel sorry about it. The paper regarding ERH was not available at that time. With review of papers, we found that ERH was related to cancer metastasis and EMT of bladder cancer [3]. According to your suggestion, we have corrected this part as follows:
The reason why we picked MKN28 cell lines is that we previously made a stable cell line successfully by injecting the rhoGDI2 gene into MKN28 cells [5].
Moreover, both introduction and materials and methods do not provide sufficient information about AIM's. Authors have to provide more detailed description of their composition (HPLC analysis?)
Thank you for your comments. According to your suggestion, we have added it.
Purification and characterization of AIM (anthocyanins in Meoru) have been described previously [6]. [7] AIMs consists of: delphinidin-3,5-diglucoside (1): cyanidin-3,5-diglucoside (2):petunidin-3,5-diglucoside (3): delphinidin-3-glucoside (4): malvdin-3,5-diglucoside (5): peonidin-3,5-diglucoside (6): cyanidin-3-glucoside (7): petunidin-3-glucoside (8): peonidin-3-glucoside (9):malvidin-3- glucoside (10) = 3.5: 3.4: 7.1: 23.9: 8.0: 9.6: 9.1: 16.1: 5.7: 13.4.
In M&M authors suddenly refers to E-MKN28 cells and M-MKN28 cells without mentioning how to interpret these differences.
Thank you for your comments. We really agree your opinion. According to your suggestion, we have added the information as follows;
We used two ERH-transfected stable MKN28 cell lines (E-MKN28 cells) in this study. For the two eoERH#5“ and „eoERH#27“ stable cell lines, they were selected in the process of making a stable cell line, and the ones with the most expression level were selected. Mock-treated MKN28 cells (M-MKN 28 cells) refer to a cell that has been transfected with a control plasmid without the ERH gene. The pCMV6 empty vector was used to control cells.
Based on the description provided by the authors the Annexin V-based experiment was performed improperly. Annexin V conjugate should be added to the cells suspended in binding buffer, not in PBS (the provided description suggests that).
Fig. 3 A and C - the axis description should be changed from caspase to pro-caspase. The present description is inaccurate and misleading.
Thank you for your comments. According to your suggestion, we have corrected it.
- How many biologically independent repeats was performed for each experiment? Whiskers of plots present SD? Some data presents almost ideal repeatability (WB results almost no discrepancies between samples?)
Thank you for your comments.
Yes, it is. The data was from the most 3 reliable results. The values were normalized against beta-actin.
minor issues
line 74 - In vitro -> in vitro
line 220 - AS -> As
line 229, 241, 265, 289, 299 x -> ×
line 275 weird symbol in 'treated' word
Thank you for your comments. According to your suggestion, we have corrected it.
- Lim, B.H.; Cho, B.I.; Kim, Y.N.; Kim, J.W.; Park, S.T.; Lee, C.W. Overexpression of nicotinamide N-methyltransferase in gastric cancer tissues and its potential post-translational modification. Exp Mol Med 2006, 38, 455-465, doi:10.1038/emm.2006.54.
- Park, J.H.; Park, M.; Park, S.Y.; Lee, Y.J.; Hong, S.C.; Jung, E.J.; Ju, Y.T.; Jeong, C.Y.; Kim, J.Y.; Ko, G.H., et al. ERH overexpression is associated with decreased cell migration and invasion and a good prognosis in gastric cancer. Transl Cancer Res 2020, 9, 5281-5291, doi:10.21037/tcr-20-1498.
- Pang, K.; Zhang, Z.; Hao, L.; Shi, Z.; Chen, B.; Zang, G.; Dong, Y.; Li, R.; Liu, Y.; Wang, J., et al. The ERH gene regulates migration and invasion in 5637 and T24 bladder cancer cells. BMC Cancer 2019, 19, 225, doi:10.1186/s12885-019-5423-9.
- !!! INVALID CITATION !!! [26].
- Cho, H.J.; Baek, K.E.; Park, S.M.; Kim, I.K.; Choi, Y.L.; Cho, H.J.; Nam, I.K.; Hwang, E.M.; Park, J.Y.; Han, J.Y., et al. RhoGDI2 expression is associated with tumor growth and malignant progression of gastric cancer. Clin Cancer Res 2009, 15, 2612-2619, doi:10.1158/1078-0432.CCR-08-2192.
- Kim, H.J.; Tsoy, I.; Park, J.M.; Chung, J.I.; Shin, S.C.; Chang, K.C. Anthocyanins from soybean seed coat inhibit the expression of TNF-α-induced genes associated with ischemia/reperfusion in endothelial cell by NF-κB-dependent pathway and reduce rat myocardial damages incurred by ischemia and reperfusion in vivo. FEBS letters 2006, 580, 1391-1397, doi:doi:10.1016/j.febslet.2006.01.062.
- Yun, J.W.; Lee, W.S.; Kim, M.J.; Lu, J.N.; Kang, M.H.; Kim, H.G.; Kim, D.C.; Choi, E.J.; Choi, J.Y.; Kim, H.G., et al. Characterization of a profile of the anthocyanins isolated from Vitis coignetiae Pulliat and their anti-invasive activity on HT-29 human colon cancer cells. Food Chem Toxicol 2010, 48, 903-909, doi:10.1016/j.fct.2009.12.031.

Reviewer 2 Report
The authors demonstrated that AIMs exert increased anti-cancer effects in ERH-transfected MKN28 cells. In particular, AIMs sustain, in ERH-transfected cells, a caspase-dependent apoptosis mediated by mitochondrial depolarization and ROS generation.
Major:
A stronger and possibly causal link between the overexpression of ERH and the increased AIMs activity should be present. To this aim:
- Since the enhancer of rudimentary homolog (ERH) gene is required for the expression of multiple cell cycle and DNA damage response (DDR) genes the authors should assess, other than the effects on cell apoptosis, also some effects on cell cycle progression (e.g., DNA synthesis by BrdU incorporation, PI staining)
- The effects of AIMs on the expression of key genes target of ERH should be addressed, this could help to evaluate whether AIMs could modulate cell viability by affecting ERH functions
- Since cancer cells driven by mutations in the KRAS oncogene are particularly sensitive to RNAi-mediated suppression of ERH function, the authors should address the functional activity of AIMs in a different cell line bearing the KRAS mutation (e.g., AGS)
Minor:
- The authors should explain and discuss whether a protein overexpressed and involved in aggressive behaviour of cancers could be responsible for a more strong anti-cancer effect of AIMs
Author Response
The authors demonstrated that AIMs exert increased anti-cancer effects in ERH-transfected MKN28 cells. In particular, AIMs sustain, in ERH-transfected cells, a caspase-dependent apoptosis mediated by mitochondrial depolarization and ROS generation.
Major:
A stronger and possibly causal link between the overexpression of ERH and the increased AIMs activity should be present. To this aim:
- Since the enhancer of rudimentary homolog (ERH) gene is required for the expression of multiple cell cycle and DNA damage response (DDR) genes the authors should assess, other than the effects on cell apoptosis, also some effects on cell cycle progression (e.g., DNA synthesis by BrdU incorporation, PI staining)Thank you for your comments. We really agree your opinion. It would be better, if the influence of ERH on cell cycle evaluation be evaluated in this study. However, although it was reported that ERH played a critical role in the mRNA splicing, the expression of the mitotic motor protein CENP-E, and consequently, chromosome congression (Fujimura et al., 2012; Weng et al., 2012), little is known about the role of ERH in cancer. In addition, although ERH is known to be related to the expression of multiple cell cycle and DNA damage response (DDR) genes, AIM did not affect the cell cycle and DNA damage. For this investigation, we should prepare a lot of things, so we will consider it in the next study. We added your good comments on the discussion parts as follows;” Thank you again for your comments.
- “Third, we did not investigate whether the increased anti-cancer effect of AIM with the increase of ERH was related to the expression of multiple cell cycle and DNA damage response (DDR) genes that is deeply involved with ERH. As it was reported, ERH played a critical role in the mRNA splicing, the expression of the mitotic motor protein CENP-E, and consequently, chromosome congression [1]. To show the strong causal relationship between the over-expression of ERH and the increased anti-cancer activity of AIMs should be present. However, the role of ERH is not fully elucidated in cancer, and AIM did not affect the cell cycle and DNA damage. To perform this investigation, we should prepare a lot of things, so we will consider it in the next study. In the next study, we will consider testing the effects of AIMs on cell cycle, nuclear morphology, chromosome congression, and the expression of multiple cell cycle and DNA damage response (DDR) genes to evaluate whether AIMs could modulate cell viability by affecting ERH functions [1].
- The effects of AIMs on the expression of key genes target of ERH should be addressed, this could help to evaluate whether AIMs could modulate cell viability by affecting ERH functions
- � Thank you for your comments. We really agree your opinion. We will consider it in the next study. We added your good comments on the discussion parts. Thank you again for your comments.
- Since cancer cells driven by mutations in the KRAS oncogene are particularly sensitive to RNAi-mediated suppression of ERH function, the authors should address the functional activity of AIMs in a different cell line bearing the KRAS mutation (e.g., AGS)
- � Thank you for your comments. We really agree your opinion. We will consider it in the next study. We added your good comments on the discussion parts. Thank you again for your comments.
Minor:
- The authors should explain and discuss whether a protein overexpressed and involved in aggressive behaviour of cancers could be responsible for a more strong anti-cancer effect of AIMs
- Thank you again for your comments.
- Although the role of ERH has not been clearly documented in cancers, Human Protein Atlas (https://www.proteinatlas.org/ENSG00000100632-ERH/pathology) suggests that ERH serves as a prognostic factor in liver, head and neck, breast, ovary, renal cell, gastric cancer, and lung cancer. However, the role is The role is reversed depending on the cancer types; High expression of ERH serves as a good prognostic factor in breast, ovary and gastric cancer while it does as a poor prognostic factors in liver, head and neck and prostate cancer. In addition, low ERH expression is associated with better survival in cancer patients whose cancers harbor KRAS mutations [1] while ERH was reportedly involved in cancer metastasis and EMT of bladder cancer [2]. Here, we investigated the effects of up-regulated ERH gene on anticancer efficacies of AIMs in terms of apoptosis by transfecting ERH gene into MKN28 human gastric cancer cells.
- Therefore, we have corrected the introduction as follows;
- Thank you for your comments. We really agree your opinion. As we know, the cancer with KRAS mutations shows aggressive and resistant to chemotherapy. Although the cancers of ERH expression is inversely associated with survival of cancer patients whose cancers harbor KRAS mutations, it is not associated with survival in those patients whose tumors are WT for KRAS or harbor EGFR. In addition, the role is The role is reversed depending on the cancer types; High expression of ERH serves as a good prognostic factor in breast, ovary and gastric cancer while it does as a poor prognostic factors in liver, head and neck and prostate cancer.
- Weng, M.T.; Lee, J.H.; Wei, S.C.; Li, Q.; Shahamatdar, S.; Hsu, D.; Schetter, A.J.; Swatkoski, S.; Mannan, P.; Garfield, S., et al. Evolutionarily conserved protein ERH controls CENP-E mRNA splicing and is required for the survival of KRAS mutant cancer cells. Proc Natl Acad Sci U S A 2012, 109, E3659-3667, doi:10.1073/pnas.1207673110.
- Pang, K.; Zhang, Z.; Hao, L.; Shi, Z.; Chen, B.; Zang, G.; Dong, Y.; Li, R.; Liu, Y.; Wang, J., et al. The ERH gene regulates migration and invasion in 5637 and T24 bladder cancer cells. BMC Cancer 2019, 19, 225, doi:10.1186/s12885-019-5423-9.

Reviewer 3 Report
The manuscript „Apoptotic effects of anthocyanins extracted from fruits of Vitis coignetiae Pulliat is enhanced by ectopical expression of en-3 hancer of the rudimentary homolog (ERH) in human gastric 4 carcinoma MKN28 cells” presents solid investigation on some molecular mechanisms included in anticancer effects of a plant extract used in traditional medicine in Asia. This is an interesting work even if it cannot answer all open questions but it could contribute to further discussions and stimulation of investigations in that field.
Therefore, the manuscript can be published after minor revision considering the following recommendations:
The title is too long and should be shortened. Generally, the authors should check throughout the manuscript for the correct use of abbreviations (all abbreviations should be explained). Some examples:
- Paragraph 2.2: Please, include here the type of cells transfected (MKN28) and explain the abbreviation for the transfected cells (E-MNK28);
- Paragraph 2.3, line 125-126: “For the trypan blue exclusion methods, the number of surviving cells was counted using trypan blue exclusion methods.” This sentence has to be rewritten including the producer of chemicals and equipment used.
- Paragraph 2.7: Please explain the meaning of the abbreviation MMP.
- Figures: The abbreviations „eoERH#5“ and „eoERH#27“ used in the figures are not explained in the main text. What is the difference in the treatment in these two kinds of transfected cells? What means “mock-treated cells”? How are they treated?
Author contribution: Please, make clear how “other authors” contributed to the research using the abbreviations of their names.
Author Response
The manuscript „Apoptotic effects of anthocyanins extracted from fruits of Vitis coignetiae Pulliat is enhanced by ectopical expression of enhancer of the rudimentary homolog (ERH) in human gastric 4 carcinoma MKN28 cells” presents solid investigation on some molecular mechanisms included in anticancer effects of a plant extract used in traditional medicine in Asia. This is an interesting work even if it cannot answer all open questions but it could contribute to further discussions and stimulation of investigations in that field.
Therefore, the manuscript can be published after minor revision considering the following recommendations:
The title is too long and should be shortened. Generally, the authors should check throughout the manuscript for the correct use of abbreviations (all abbreviations should be explained). Some examples:
Thank you for your comments. We really agree your opinion. According to your suggestion, we have corrected it.
Paragraph 2.2: Please, include here the type of cells transfected (MKN28) and explain the abbreviation for the transfected cells (E-MNK28);
Thank you for your comments. We really agree your opinion. According to your suggestion, we have corrected it.
Paragraph 2.3, line 125-126: “For the trypan blue exclusion methods, the number of surviving cells was counted using trypan blue exclusion methods.” This sentence has to be rewritten including the producer of chemicals and equipment used.
Thank you for your comments. We really agree your opinion. According to your suggestion, we have corrected it.
For the trypan blue exclusion methods, the supernatant was discarded, and the pellet was resuspended with culture medium. The cell suspension was mixed with trypan blue dye (Sigma-aldrich Chemical Co.), transferred to a hemocytometer, and live cells were counted under a phase-contrast microscope (Carl Zeiss, Oberkochen, Germany)
Paragraph 2.7: Please explain the meaning of the abbreviation MMP.
Thank you for your comments. According to your suggestion, we have added the information in the manuscript..
mitochondrial membrane potential (MMP, ΔΨm)
Figures: The abbreviations „eoERH#5“ and „eoERH#27“ used in the figures are not explained in the main text. What is the difference in the treatment in these two kinds of transfected cells? What means “mock-treated cells”? How are they treated?
Thank you for your comments. We really agree your opinion.
According to your suggestion, we have added the information in the manuscript..
è
We used two ERH-transfected stable MKN28 cell lines (E-MKN28 cells) in this study. For the two eoERH#5“ and „eoERH#27“ stable cell lines, they were selected in the process of making a stable cell line, and the ones with the most expression level were selected. Mock-treated MKN28 cells (M-MKN 28 cells) refer to a cell that has been transfected with a control plasmid without the ERH gene. The pCMV6 empty vector was used to control cells.
Author contribution: Please, make clear how “other authors” contributed to the research using the abbreviations of their names.
Thank you for your comments. We really agree your opinion. According to your suggestion, we have corrected it.
S.G., S.J., J.Y., H.C., Y.L., H. K., S. L., H. J. K., and G.S.K. reviewed the manuscript and gave comments on the results and the manuscript.

Reviewer 4 Report
Authors in this paper verified the anticancer efficacy of AIMs in ERH-transfected MKN28 cells (E-MKN28 cells). ERH was previously identified as differential expressed protein between normal and cancer tissues and for this reason chosen. The paper should be very interesting for readers. Unfortunatly the lacking of a biological role of ERH in GC makes the results ineffective.
Major concern:
In the discussion section Authors declared that “the role of ERH is not fully documented in GC and this represent a weakness of this study. Moreover, authors declared that ERH can enhance invasion and it inhibited cancer cell migration and invasion (data not shown). In addition, immunohistochemical staining with tissue microarray harboring tissue samples showed that a lower cancer recurrence rates and longer survival times than patients tumors showing low expression of ERH (p = 0.04, 359 data not shown). This finding suggests that ERH may serve as a good prognostic factor. However, this result is opposite to previous reports in other cancers (ref).
It is opinion of this reviewer that these “not shown results” are necessary to justify the founded results. Infact, very interesting results are already acquired by authors, so it is sufficient to reshape the layout of the paper to make the results much more robust.
So I suggest to show the role of ERH transfection in GC cells and than the effect of AIMs treatments.
Minor concern:
1) The western blotting analysis showing the Upregulation of EHR in transfected cells should be added
2) In MM the section regarding the extraction method and characterization of AIMs is lacking
3) Construction of the RhoGDI2 expression plasmid and transfection. RhoGDI2 is the HER gene? Please check
4) in the discussion section please add the ref in the sentence "this result is opposite to previous reports in other cancers (ref)".
Author Response
Authors in this paper verified the anticancer efficacy of AIMs in ERH-transfected MKN28 cells (E-MKN28 cells). ERH was previously identified as differential expressed protein between normal and cancer tissues and for this reason chosen. The paper should be very interesting for readers. Unfortunatly the lacking of a biological role of ERH in GC makes the results ineffective.
Major concern:
In the discussion section Authors declared that “the role of ERH is not fully documented in GC and this represent a weakness of this study. Moreover, authors declared that ERH can enhance invasion and it inhibited cancer cell migration and invasion (data not shown). In addition, immunohistochemical staining with tissue microarray harboring tissue samples showed that a lower cancer recurrence rates and longer survival times than patients tumors showing low expression of ERH (p = 0.04, 359 data not shown). This finding suggests that ERH may serve as a good prognostic factor. However, this result is opposite to previous reports in other cancers (ref).
Thank you for your comments. We really agree your opinion.
According to your suggestion, we have added the information in the manuscript..
The investigation was performed some of our team. They recently published the paper. I added the reference.
It is opinion of this reviewer that these “not shown results” are necessary to justify the founded results. Infact, very interesting results are already acquired by authors, so it is sufficient to reshape the layout of the paper to make the results much more robust.
Thank you for your comments. We really agree your opinion.
According to your suggestion, we have added the information in the manuscript..
Our team recently published the paper. I added the reference.
So I suggest to show the role of ERH transfection in GC cells and than the effect of AIMs treatments.
Thank you for your comments. We really agree your opinion. The investigation was performed some of our team. Now, the paper was published, I added the reference.
Minor concern:
1) The western blotting analysis showing the Upregulation of EHR in transfected cells should be added
Thank you for your comments. We really agree your opinion. The investigation was performed some of our team. Now, the paper was published, I added the reference.
2) In MM the section regarding the extraction method and characterization of AIMs is lacking
Thank you for your comments. According to your suggestion, we have added the information in the manuscript..
3) Construction of the RhoGDI2 expression plasmid and transfection. RhoGDI2 is the HER gene? Please check
Thank you for your comments. We are sorry that we did not notice the mistakes. We have corrected the information properly.
4) in the discussion section please add the ref in the sentence "this result is opposite to previous reports in other cancers (ref)".
Thank you for your comments. According to your suggestion, we have added the information in the manuscript..

Round 2
Reviewer 1 Report
I don't have further comments for authors. All of the raised issues were addressed and/or corrected.
Reviewer 2 Report
No experiments accordingly to reviewer's comments have been performed, only comments to the discussion section have been added.
Reviewer 4 Report
Authors addressed the reviewer's doubts. However, for the next time, I recommend the authors to be more honest and write “data not shown” only if the results are missing from the paper and write “in press” or “submitted to other journal” if the results are the subject of other paper!